# Adenomyosis as a Risk Factor for Myometrial or Endometrial Neoplasms—Review

**DOI:** 10.3390/ijerph19042294

**Published:** 2022-02-17

**Authors:** Maria Szubert, Edward Kozirog, Jacek Wilczynski

**Affiliations:** Clinic of Surgical and Oncologic Gynecology, 1st Department of Gynecology and Obstetrics, Medical University of Lodz, M. Pirogow’s Teaching Hospital, Wilenska 37 St., 94-029 Lodz, Poland; edward.kozirog@gmail.com (E.K.); jrwil@post.pl (J.W.)

**Keywords:** adenomyosis, endometrial cancer, malignant transformation of adenomyosis

## Abstract

Adenomyosis is a common benign gynecological condition, defined as an extension of endometrial tissue into the myometrium. Some studies suggest that adenomyosis could be a favorable prediction factor associated with survival outcomes in endometrial cancer. The aim of our systematic review was to investigate the current knowledge regarding adenomyosis and a possible molecular mechanism of carcinogenesis in adenomyotic lesions. In addition, the long-term prognosis for patients with endometrial cancer and coexisting adenomyosis (and endometriosis) was a key point of the research. The current literature was reviewed by searching PubMed, using the following phrases: “adenomyosis and endometrial cancer” and “malignant transformation of adenomyosis”. According to the literature, genetic mutations, epigenetic changes, and inactivation of specific tumor suppressor genes in adenomyosis are still poorly understood. Data regarding the influence of adenomyosis on survival outcomes in endometrial cancer seem to be contradictory and require further clinical and molecular investigation.

## 1. Introduction

Adenomyosis, endometriosis, and gynecological cancers, such as endometrial, ovarian endometroid, and clear cell cancers, may have common pathogenetic mechanisms that, amongst others, include hormonal factors, genetic predisposition growth factors, inflammation, altered function of the immune system, environmental factors, and oxidative stress. Adenomyosis is described as the presence of both endometrial epithelium and stroma within the muscle layer of the uterus [1,2]. It causes symptoms such as irregular bleeding, spotting, painful menses, and infertility. There are two forms of adenomyosis—diffuse and focal, usually identified during trans-vaginal ultrasound (US) examination [3]. Van der Bosch et al. modified the Morphological Uterus Sonographic Assessment (MUSA) group consensus in order to implement and organize the sonographic classification and reporting system for the diagnosis of adenomyosis (but it has not yet been implemented worldwide) [4]. The general diagnostic US features of two forms of adenomyosis are presented in Table 1. [2].

However, ultrasound findings in adenomyosis may sometimes be confounding because of pathological reasons. While it is still difficult to establish the diagnosis of adenomyosis based only on ultrasound, the possible interconnections between adenomyosis and endometrial cancer (EC) make the diagnosis of EC in an adenomyotic uterus even more difficult. Adenomyosis can coexist with EC, or cancer may spread from the endometrium into adenomyotic foci or vice versa [5]. There are no pathognomic features of EC in adenomyosis that could be detected in ultrasound. While an irregular endometrium layer or junctional zone typical of adenomyosis may imitate endometrial cancer, thickening of the myometrium may be a symptom for both adenomyosis and the spread of cancer. These confounding features may lead, in our opinion, to the early recognition of EC associated with adenomyosis.

In this review, we would like to systemize the current knowledge about different types of interconnections between adenomyosis and endometrial cancer or other cancers of women’s reproductive systems. We also studied possible molecular mechanisms of carcinogenesis in adenomyotic lesions. In addition, the long-term prognosis for patients with endometrial cancer and coexisting adenomyosis (and endometriosis) was a key point of this research.

We performed advanced searches for PubMed literature using the phrases: “adenomyosis and endometrial cancer” and “malignant transformation of adenomyosis”. From 545 citations covering the period 1954–2020, 470 articles were initially excluded after title, abstract or content screening by two independent authors. The next 26 articles were excluded after a full text screening and during data collection. Finally, 39 articles were included for the analysis. Details of the results retrieved are shown according to the PRISMA 2020 guidelines on Figure 1 [6].

## 2. Adenomyosis and Risk of Malignant Transformation

### 2.1. Established Pathological Pathways in Adenomyosis

It has been observed already that adenomyosis is associated with EC. However, the process of malignant transformation in adenomyosis remains unclear [7]. Adenomyosis coexists with other estrogen-dependent benign gynecological conditions, such as endometrial polyps or myomas. It may be suggested that high levels of estrogen play a role in the development of adenomyosis and endometrial cancer [8,9]. Estrogens are found to interact on phospholipid membranes through annexins [10]. The annexin A2 (ANXA2) is a crucial molecule in the activation of metastasis, angiogenesis, and endometrial tissue growth. Annexins belong to the group of estrogen-responsive proteins in eutopic endometrium. In adenomyosis, the expression of ANXA2 is upregulated. It has been noticed that the overexpression of ANXA2 is correlated with markers of the epithelial to mesenchymal transition. These molecules (ANXA2) can play a dual role in the pathogenesis of adenomyosis: first, through spreading potential and, second, through angiogenic capacity [11]. The expression of ANXA2 in cells is upregulated in endometrial carcinoma compared to endometrial tissue (95.2%/55.6%; *p* < 0.05). The strong correlation between the expression of ANXA2 and the FIGO stage, degree of differentiation, myometrial invasion, and lymph node metastasis has already been reported. Lu Deng et al. documented that the overexpression of ANXA2 in EC is an independent risk factor for poor prognosis (*p* < 0.05, hazard ratio [HR] = 8.004) [12]. The mutation of p53 genes was investigated in EC. A correlation between a p53 mutation in EC and clinical outcomes has already been confirmed. The presented data showed that the dominant-negative p53 mutation was associated with advanced stages (*p* = 0.01), nonendometrioid-type tumors (*p* = 0.01), and grade 3 tumors (*p* = 0.04). In relation to patients without mutation (with wild type) and those with a recessive mutation in the p53 gene, patients with a dominant-negative mutation had significantly shorter survival. In addition, dominant-negative p53 mutation occurred as the most important prognostic factor for stage III/IV endometrial cancer (*p* = 0.0023) [13]. Finally, the mutation of the p53 protein was detected in hyperplastic and atypical epithelium of carcinoma arising from adenomyosis foci [14].

The collected data suggest that altered molecular pathways that play a role in promoting adenomyosis and endometrial cancer are common. Increased angiogenesis, abnormal tissue growth, and invasion ability occur in both. In both conditions, there are observed changes in the microenvironment, such us: a high level of vascular endothelial growth factor, increased production of reactive species of oxygen and pro-inflammatory cytokines, KRAS (V-Ki-ras2 Kirsten rat sarcoma viral oncogene homolog; protooncogene responsible for growth factors) mutations, and the same, but smaller-scale, progesterone resistance, epithelial mesenchymal transition, and fibroblast-to-myofibroblast trans-differentiation. A series of next-generation sequencing (NGS) studies give the opportunity to distinguish the cellular origins of adenomyosis and highlight pathogenetic differences between other benign conditions, such as endometriosis and fibroids. Variations in the presence of recurrent specific mutations in this disease allow us to reach the conclusions that (1) driver mutations found in smooth muscle cells of uterine fibroids are absent in adenomyosis and (2) KRAS and other less frequent mutations are limited to endometrial-type epithelial cells [15].

### 2.2. Adenomyosis as an Oligoclonal Disorder Strongly Associated with KRAS Mutation

The molecular pathogenesis of adenomyosis still remains unclear. Inoune et al. documented recurrent KRAS mutation in adenomyosis in about 37% of adenomyosis cases. The KRAS protein is responsible for activating groups of growth factor proteins, as well as other cell-signaling receptors. In the case of KRAS-mutated adenomyotic clones, the sensitivity of dienogest treatment was decreased significantly. Dienogest is one of the most extensively studied progestins in endometriosis and adenomyosis. Furthermore, a decrease in PR (progesterone receptor) protein levels in the epithelial components was observed as well. These findings suggest that adenomyotic lesions that contain KRAS mutation may contribute to reduced dienogest treatment efficacy by the suppression of PR expression. This genetic alteration could be relevant to evaluate the potential relapse risk in patients on progestins therapy or/with concomitant surgical treatment [16]. Similar observations on the presence of KRAS mutation and sensitivity to cetuximab (monoclonal antibody against epidermal growth factor receptor EGFR) in colorectal cancer have already been implemented into clinical practice [17]. In addition, a needle biopsy of an adenomyosis lesion with the assessment of KRAS mutation status and/or PR expression might be a valuable diagnostic possibility and, in future, in the case of KRAS-mutated lesion genetically guided therapy.

### 2.3. Possible Interactions between Adenomyosis, Endometriosis and Gynecological Cancers

Recurrent mutations, such as KRAS, PIK3CA (phosphatidylinositol-4,5-bisphosphate 3-kinase), PPP2R1A (Protein Phosphatase 2 Scaffold Subunit Alpha), and ARID1A (AT-Rich Interaction Domain 1A), are observed in endometriosis and adenomyosis, which suggests similar disease development. None of the above is detected in uterine fibroids, which suggests another molecular pathway in developing uterine fibroids [15]. In view of the above-described fact, one should assume that adenomyosis and endometriosis could have common molecular pathways into carcinogenesis. Malignant transformation in endometriosis occurs in two ways: first, transformation of atypical lesions of ovarian endometriomas, which are 60–80% of cases of EOAC (endometriosis associated ovarian cancer); and second, squamous and mucinous metaplasia arising from endometriosis. Gounaris et al. suggest that mutations, such as PTEN (Phosphatase and Tensin Homolog), ARD1A, PIK3CA- mTOR (phosphoinositide 3-kinases—mammalian target of rapamycin), and Ras-Raf-MAPK pathway activation in eutopic endometrium predispose one to the development of endometriosis tissue. These genetic alterations are described as a “bad endometrium”, which may promote malignant transformation (arising out of endometriosis) into EAOC, in the case of women with retrograde menstruation [18]. In adenomyosis and endometriosis, genetic alterations leading to malignant transformation have been proven in vivo. A very interesting case was described by Santoro et al., where a patient operated on for pelvic mass was, in the final pathologic examination, diagnosed for endometriosis, clear cell carcinoma of the ovary, adenomyosis, and stromal sarcoma of the uterus, simultaneously [19]. All genetic alterations discovered up to the present day that play a role in the malignant transformation in endometriosis and adenomyosis are described in Table 2.

Apart from genetic mutations, which may promote carcinogenesis, local chronic inflammation typical for endometriosis and adenomyosis induces angiogenesis, cell proliferation, the inhibition of apoptosis, production of ROS that enhances DNA damage, and mutation [33,34]. Inflammatory factors associated with endometriosis and gynecological cancers are presented in Table 3. One must pay attention to the fact that studies on inflammatory factors mainly describe peritoneal endometriosis and not adenomyosis itself.

### 2.4. Adenomyosis Originating from the Invasion and Migration of the Endometrium

Spreading of adenomyosis depends on cancer-, cell motility and inflammation- (CMI)-associated terms, cell proliferation, and angiogenesis. Cells subunits with high copy number variation (CNV) levels possessing tumor-like features were confirmed through single-cell RNA sequencing (scRNA-seq). Liu et al. proved the theory about the migration and invasion of the endometrium into the myometrium. Their results indicated that the inhibition of EET (epithelial–endothelial transition) and VM formation (vascular-mimicry formation) may be a potential strategy for adenomyosis management [42].

## 3. Clinical Studies

### 3.1. Adenomyosis as an Oncological Prognostic Marker in Endometrial Cancer—FIGO Stage and Grade

Data suggest that the coexistence of adenomyosis and adenocarcinoma in patients is a confirmed favorable factor; however, the exact rate of coexistence is not well-recognized in epidemiological studies. Patients with EC coaffected by adenomyosis have low histologic tumor grades and a better prognosis. There is an observed, significantly lower frequency of metastasis into the lympho-vascular space and lymph nodes in postoperative histological examination. What is more, the incidence of adenomyosis and EC is associated with a history of using estrogen-based complex hormonal therapy. In this case, endometrioid tumors are well differentiated. It gives an opportunity for early diagnosis while the tumor would be still confined to the uterus [7]. On the contrary, according to Ismiil N et al., adenomyosis is a significant risk factor for deep myometrial invasion. In the case of grade FIGO I endometrial endometrioid adenocarcinoma, deep myometrial invasion was observed more frequently in patients affected by adenomyosis (91.3% to 63.8%), probably through enlarging the surface area of its interface with adhering myometrium. Myometrial invasion was examined by C10-negative staining around glands with a jagged outline surrounded by inflamed desmoplastic stroma [43]. Gizzo et al. performed a retrospective analysis of 289 patients diagnosed with endometrial cancer who underwent a total hysterectomy with concomitant bilateral salpingo-oophorectomy and pelvic-lymphadenectomy. Adenomyosis was associated with a lower FIGO stage. In patients with coexisting adenomyosis, FIGO stage I was assigned in 83.8% vs. 68.7% without adenomyosis (*p* < 0.01). In addition, borderline statistically significant differences were found in the tumor grade. In Gizzo’s study, adenomyosis was highly associated with the following factors: diabetes, hypertension, high BMI, and tamoxifen intake [44], the same risk factors as for type I EC, according to the Bokham classification. This study showed that adenomyosis is a favorable prognostic marker in patients with endometrial cancer.

### 3.2. Prevalence of Adenomyosis in Gynecological Cancers Other Than Endometroid Endometrial Cancer

There are also several case reports on sarcomas arising in adenomyosis foci described in the literature [45,46,47]. Malignant transformation in adenomyosis other than endometroid endometrial cancer has been documented [48]. Bingjian Lu et al. describe three cases of serous carcinoma arising from uterine adenomyosis or an adenomyotic cyst of the cervical stump [49]. In the case of a woman with a history of tamoxifen therapy, papillary serous carcinoma arising from adenomyotic foci has been reported [50]. J I Choi et al. reported the rapid appearance of a low-grade endometrial stromal sarcoma (ESS) after uterine fibroid embolization for presumed adenomyosis [51]. It has also been documented that disseminated intraperitoneal ESS (bowel and liver parenchymal metastasis) can be a consequence of supracervical hysterectomy with morcellation uteri due to adenomyosis [52]. According to Talia et al., the current FIGO staging system for uterine adenosarcoma assumes origin from the surface endometrium and does not address the rare occurrence of intramural tumors that are connected with adenomyosis [46].

### 3.3. Adenomyosis and Endometrial Cancer as Two Different, Independent Entities—Influence on Survival

Adenomyosis and endometrial cancer share common microenvironment and etiopathogenetic mechanisms that favor cell proliferation and inflammation in the uterus. Several studies implied the significance of existing adenomyosis in myometrium as a potential risk factor for endometrial cancer originating in eutopic endometrium (so called EC-A—endometrial cancer with associated adenomyosis) [53,54]. Hermens et al. performed a population-based retrospective cohort study of almost 130 thousand patients with endometriosis/adenomyosis in Denmark, matched with a comparison group of women with a nevus (according to pathology registry from 1990 to 2015). In his study, EC was significantly more often diagnosed in the endometriosis/adenomyosis group. In about 20% of EC cases, the diagnosis was established at random, only because a hysterectomy due to adenomyosis had been performed. It may support our theory that careful ultrasound screening for adenomyosis in AUB (abnormal uterine bleeding) patients may be helpful in the early diagnosis of EC [54].

The first systematic review and meta-analysis on the prevalence of adenomyosis in endometrial cancer patients performed by Raffone et al. presented a lack of relation between adenomyosis and endometrial cancer. In the Raffone’s group, the pooled prevalence of adenomyosis was 22.6%, which was approximate to that reported for myomas. In this meta-analysis, worth mentioning is also the lack of differences in the prevalence of EC histotype and FIGO, regarding patients with or without adenomyosis, which, in Raffone’s opinion, does not confirm the hypothesis that adenomyosis may be a risk factor for endometrial cancer [55]. In a systemic review published by Diego Raimondo et al., the prognoses of endometrial cancer patients with and without coexisting adenomyosis were compared. EC patients with coexisting adenomyosis had half the risk of death and recurrence compared to women without adenomyosis in an univariate analysis. However, in a multivariate analysis, the risk of EC recurrence in patients with adenomyosis seemed to be irrelevant. Finally, in the case of overall survival, a multivariate analysis was not performed [56].

Conversely, An M et al., in their meta-analysis, also indicated that coexistent adenomyosis with endometrial cancer (EC-A) is associated with a favorable prognosis compared to EC. They found a better overall survival rate in EC-A, decreased ratio of Ib according to FIGO (deep myometrial invasion) (OR = 0.45; 95% CI = 0.33–0.60; *p* < 0.00001) or lymphovascular space invasion [57]. Worth mentioning is the fact that only parity and not other risk factors for EC differs between patients with EC coexisting with adenomyosis and EC alone [58].

### 3.4. Direct Malignant Transformation of Adenomyosis Foci into Endometrial Cancer

The literature also provides evidence of direct malignant transformation of adenomyosis as Endometrial Cancer Arising In Adenomyosis (EC-AIA). EC-AIA, which accounts for less than 1% of EC, should be diagnosed only if some important histopathological features are acknowledged: EC must not be present in the eutopic endometrium or other places in the pelvis; cancer must arise from the epithelium of adenomyotic foci found between the uterus muscle; and the diagnosis of adenomyosis should be confirmed by the presence of endometrial stromal cells surrounding the ectopic endometrial glands. A comparison of EC-AIA and EC-A (cancer coexisting with adenomyosis) leads to the conclusion that endometrial cancer arising in adenomyosis is associated with poor survival outcomes compared to EC-A. EC-AIA is likely to present aggressive tumor features (non-endometrioid histology, deep myometrial invasion, and sometimes a sarcomatous component). Possible relationships between adenomyosis and endometrial cancer are presented in Figure 2.

Mahida H et al., in multivariate and univariate analyses, demonstrated shorter disease-free survival (DFS) in patients with EC-AIA. In their study, EC-AIA remained an independent negative prognostic factor associated with decreased DFS compared to EC-A (adjusted-hazard ratio: 2.87; 95% confidence interval: 1.44–5.70; *p* = 0.031) [53]. What is more, it has been noticed that adenocarcinoma that arise from adenomyosis uteri could be present in various histological appearances [59]. Mahmoud K et al. proposed creating other clinical approaches for staging cancers where myoinvasion is found deep in myometrium [60]. Well described EC-A and EC-AIA pathological pathways do not explain coexistence of both these entities (see red line in Figure 2).

## 4. Conclusions

Adenomyosis has been frequently observed in hysterectomy specimens for endometrial adenocarcinoma. In this review, we acknowledged that adenomyosis may be a potential risk factor for myometrial or endometrial neoplasms. Adenomyosis and endometriosis are known as estrogen-dependent benign diseases, with common molecular pathways. Both these conditions may undergo a malignant transformation. However, in the case of endometriosis, the possible molecular mechanisms of carcinogenesis are well described. Carcinomas derived from endometriosis have been well established; histological subtype, epidemiologic EAOC risk was identified. On the contrary, the neoplastic potential in adenomyosis is poorly understood, and has been mainly described in association with EC. The coincidence between these two diseases can be an effect of potential common risk factors, which may lead to hyper-estrogenic status (history of diabetes, hypertension, high BMI, and tamoxifen intake, but not nulliparity [58]) or by shared pathogenetic mechanisms, including genetic mutation and inflammatory factors that induce angiogenesis and cell proliferation, which promote carcinogenesis. According to the presented data, adenomyosis seems to be a favorable prognostic oncological factor when it occurs with EC, but the finding appears to be contradictory. EC-AIA is a rare phenomenon, but it is associated with higher malignancy and nodal metastasis risk. It is necessary to validate the process of detection of adenomyosis in an intraoperative frozen section for assessing the necessity for lymphadenectomy. Considering this, more caution may be required when selecting the most appropriate histopathological confirmation of adenomyosis and surgical treatment for patients with hyper-estrogenic status and with risk factors, regardless of potential malignant transformations in adenomyosis. In clinical practice, early, more effective recognition and precise standards in ultrasonography to identify adenomyosis are required. Implementing MUSA standards could lead to higher diagnosis rates of adenomyosis. The available data do not allow us to evaluate the immediate influence of adenomyosis on EC and we are still unable to predict which women with adenomyosis will develop EC.

Further, prospective large-scale cohort studies investigating the risk of malignant transformation of adenomyosis are needed. Prospective studies focused on the life risk of malignant transformation should be undertaken to prove which women should undergo a definitive treatment of adenomyosis. There is a need to elaborate the “decision making tree” in patients with adenomyosis towards radical treatment, especially in patients with hormone-replacement therapy or endocrine treatment of breast cancer or other risk factors of EC, such as obesity, diabetes, and hypertension.

In the future, algorithms including all available prognostic values—genetic mutation, in this respect KRAS mutation as a factor for efficacy of hormonal therapy; ultrasound features (depth of myometrial invasion); and clinical characteristics (age, concomitant disease, BMI)—might lead to the individualized management of patients with adenomyosis, with a possible malignant transformation of adenomyosis into cancer.

## Figures and Tables

**Figure 1 ijerph-19-02294-f001:**
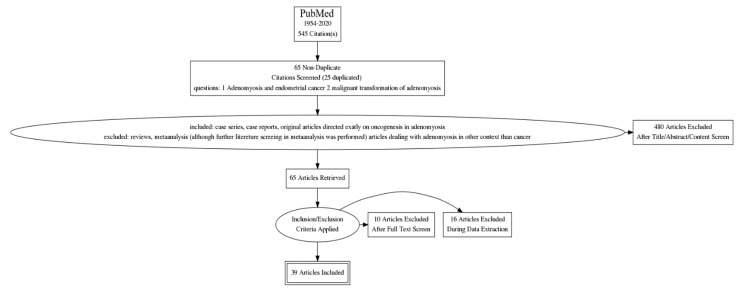
Searching strategy – flow chart[((“adenomyosis OR “adenomyosis” [All Fields] OR “adenomyoses” [All Fields]) AND (“endometrial neoplasms” OR (“endometrial” [All Fields] AND “neoplasms” [All Fields]) OR “endometrial neoplasms” [All Fields] OR (“endometrial” [All Fields] AND “cancer” [All Fields]) OR “endometrial cancer” [All Fields])) AND (1954:2020[pdat]) and (“malign” [All Fields] OR “malignance” [All Fields] OR “malignances” [All Fields] OR “malignant” [All Fields] OR “malignants” [All Fields] OR “malignities” [All Fields] OR “malignity” [All Fields] OR “malignization” [All Fields] OR “malignized” [All Fields] OR “maligns” [All Fields] OR “neoplasms” OR “neoplasms” [All Fields] OR “malignancies” [All Fields] OR “malignancy” [All Fields]) AND (“transform”[All Fields] OR “transformability” [All Fields] OR “transformable” [All Fields] OR “transformant”[All Fields] OR “transformants” [All Fields] OR “transformation” [All Fields] OR “transformations” [All Fields] OR “transformed” [All Fields] OR “transforms” [All Fields]) AND (“adenomyosis” OR “adenomyosis”[All Fields] OR “adenomyoses”[All Fields]).

**Figure 2 ijerph-19-02294-f002:**
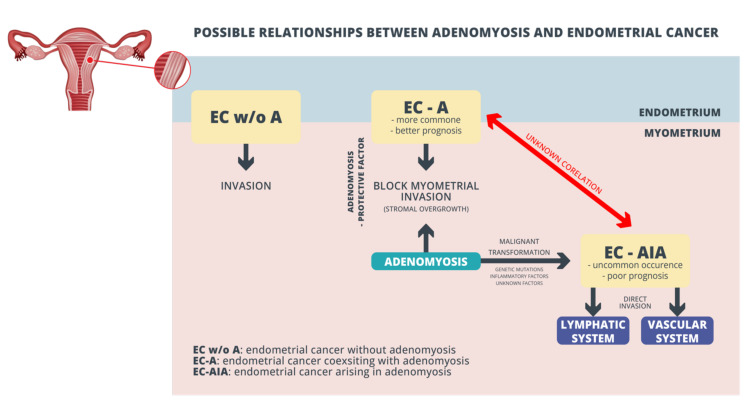
Possible relationships between adenomyosis and endometrial cancer.

**Table 1 ijerph-19-02294-t001:** Sonographic features of diffuse and focal adenomyosis (own modifications according to MUSA guidelines).

ADENOMYOSIS IN ULTRASOUND
globally enlarged uterus
asymmetric thickness anterior and posterior wall = pseudo-widening sign
cystic myometrium (cystic anechoic spaces)
junctional zone (JZ) not clearly visible, JZ interrupted, irregular, thickened; with anechoic cysts, hyperechoic dots
heterogeneous echogenicity of the myometrium
ill-defined lesion (difficult to delineate)
focal disturbances in myometrium layer
sometimes focal form diagnosed as intramural myoma
anechoic cysts or cysts of ground-glass appearance
absence of blood flow in lesions

**Table 2 ijerph-19-02294-t002:** Genetic mutations in endometriosis and adenomyosis.

Genetic Factors	Function	Association
**Laminin-5 gamma2 chain**	membrane glycoprotein, ligand of various transmembrane receptors	overexpressed in clear cell adenocarcinoma arising from adenomyosis [20]
**β-Catenin (CTNNB1)**	plasma membrane, responsibility for cell differentiation	β-Catenin pathways are involved in endometriosis and endometrial cancer, play important role in the pathogenesis of adenomyosis through epithelial–mesenchymal transition [21]
**AT-Rich Interaction Domain 1A (ARID1A)**	suppressor gene	occurrence in endometriosis tissues, mutation and loss of function in EAOC [22]
**PIK3CA (phosphatidylinositol-4,5-bisphosphate 3-kinase)**	suppressor gene	upregulated expression in adenomyosis, frequently observed in EC [23], detected in precursor endometriosis tissues, strongly associated with Ovarian Clear Cell Carcinoma (CCC) [22]
**Phosphatase and Tensin Homolog (PTEN gene)**	suppressor gene	occurrence in endometrial cyst, inactivation in up to 40% of clear carcinoma cells [24]
**Protein 53 (p53)**	suppressor gene	detected in hyperplastic and atypical epithelium of carcinoma arising from adenomyosis foci [12], mutation and loss of function in ovarian cancer [24], not observed in endometriosis, observed in endometriotic cells located near ovarian cancer cells [25]
**Wilms tumor suppressor gene (WT1)**	regulates the expression of insulin growth factor IGF-1, and transforming growth factor associated with DNA mismatch repair system	significant downregulated in endometriosis (downregulation of WT1 increased level of P450 aromatase expression and estrogen formation in endometriosis) [26], correlated with high-grade serous ovarian carcinomas [27]
**KRAS genes**	oncogene	strongly associated with adenomyosis, detected in ovarian clear cell cancer, detected in atypical endometriosis [28]
**Hepatocyte nuclear factor (HNF—1B)**	oncogene, plays role in chemoresistance	detected in endometriosis and clear cell carcinoma [29]
**Hypermetylation of MutL Homolog 1** **(MLH1)**	component of DNA mismatch repair, leading to PTEN dysfunctionMicro-satellite instability (MSI)	observed in epithelial ovarian cancer and endometriosis [30], observed in ovarian cancer at chromosome 10q23 region [31]
**Mucin 1—transmembrane hetemrodimer molecules** **(MUC1)**	member of the mucin family molecules	present in endometriotic lesions and overexpressed in epithelial ovarian tumors [32]

**Table 3 ijerph-19-02294-t003:** Inflammatory factors connected with gynecological cancers.

Inflammatory Factors	Function	Association
**Cyclooxygenase-2 (COX-2)**	promotion of angiogenesis in adenomyosis	upregulated expression in endometrial, ovarian, and cervical cancer [35,36]
**Tumor necrosis- alfa** **(TNF alfa)**	promote production of ROS	high level in endometriosis and ovarian cancer [37]
**Toll-like receptors (TLRs)—intracellular signaling components**	cell surface sensors initiate proliferation and modulate immune cells—connected with chemoresistance	well described in endometriosis and ovarian cancer [38]
**Tumor associated macrophages (TAMS)**	promote angiogenesis, tumorigenesis, matrix remodeling, inhibits adaptive immunity	infiltrated ovarian tumor [39]
**Nuclear factor kappa-light-chain-enhancer of activated B cells (NF-KB proteins)**	active in tumor cells, mediate metastasis	involved in development of endometriosis and ovarian cancer [40]
**Macrophage migration inhibitory factor (MIF protein)**	regulator of immune and inflammatory response	found in active ectopic endometrial implants [41]

## Data Availability

Not applicable.

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
