# Peer review of "Adenomyosis as a Risk Factor for Myometrial or Endometrial Neoplasms—Review"

_ijerph, 2022, doi:10.3390/ijerph19042294_

Round 1
Reviewer 1 Report
1. I congratulate authors for the proposed study. The study is well done and the reading is pleasant.
2. However is the clinical profile of endometrial cancer patients with adenomyosis different from the clinical profile of EC patients without adenomyosis? Since risk factors of endometrial cancer, such as obesity, hypertension, nulliparity are increasing, please include published studies evaluating this feature.
3. FIGO grade and stage, histotype, deep myometrial infiltration and lymphovascular space invasion, are well-known histological prognostic factors of endometrial cancer. Please describe these prognostic factors reporting studies assessing their prevalence in the population of endometrial cancer patients with adenomyosis.
4. I suggest using an abbreviation for "endometrial cancer".
Author Response
Dear Reviewer,
thank you for your valuable comments and appreciating words,
below I described changes made according to your suggestions:
- I congratulate authors for the proposed study. The study is well done and the reading is pleasant.
- However is the clinical profile of endometrial cancer patients with adenomyosis different from the clinical profile of EC patients without adenomyosis? Since risk factors of endometrial cancer, such as obesity, hypertension, nulliparity are increasing, please include published studies evaluating this feature.
Dear Reviewer, thank you for your comments. We did not find any differences exept in parity between EC and EC – A patients. We updated literature and chapter 3 accordingly (see lines 264-268).
- FIGO grade and stage, histotype, deep myometrial infiltration and lymphovascular space invasion, are well-known histological prognostic factors of endometrial cancer. Please describe these prognostic factors reporting studies assessing their prevalence in the population of endometrial cancer patients with adenomyosis.
Answer: These studies were described in Chapter 3 – “Adenomyosis and Endometrial Cancer as two different, independent entities – influence on survival” and „Adenomyosis as an oncological prognostic marker in Endometrial Cancer – FIGO stage and grade” - I suggest using an abbreviation for "endometrial cancer". – used throughout the text.,
Reviewer 2 Report
ijerph-1576214-peer-review
Review: Harald Krentel, MD – 26h of January 2022
Thank you for inviting me as reviewer of this well-written review.
1.Your overall opinion of the manuscript.
Very good review. Highly important topic.
- Your recommendations, with reasons.
Publish after minor revision.
- Comments
- Line 31: Instead of US – the complete word ultrasound should be used
- Line 29: a lot of symptoms does not seem a very scientific language
- Table 1: The reliability of different ultrasound parameters in adenomyosis remains unclear. The differentiation in table 1 is not clear. Anechoic cysts might also appear in diffuse adenomyosis. A focal adenomyosis can also cause enlargement. I would suggest to better just present a table of the ultrasound signs in adenomyosis in general.
- Line 42: The diagnosis of adenomyosis by TVS is still difficult. The lines 42 – 46 suggest that EC in adenomyosis is a common finding. But the contrary is true. EC in patients with adenomyosis is rare. And there is no literature on the differentiation of transformation or coexistence by TVS. Thus, these lines could be discussed in the discussion section.
- Line 309: I agree. However, I think the difficulty to detect EC in patients with adenomyosis should be highlighted. The typical early uterine bleeding might be missing or might be misunderstood. The sonographic appearance of EC in an adenomyotic uterus might not be visible. What does this mean for clinical counselling? Maybe it should be discussed that this could also influence the decision making in patients with adenomyosis towards hysterectomy, especially in patients with hormone-replacement therapy or endocrine treatment of breast cancer
Author Response
Dear Reviewer,
thank you very much for your valuable comments which helped us to improve our manuscript. I would like to describe our changes in details below:
- Line 31: Instead of US – the complete word ultrasound should be used - changed
- Line 29: a lot of symptoms does not seem a very scientific language - changed
- Table 1: The reliability of different ultrasound parameters in adenomyosis remains unclear. The differentiation in table 1 is not clear. Anechoic cysts might also appear in diffuse adenomyosis. A focal adenomyosis can also cause enlargement. I would suggest to better just present a table of the ultrasound signs in adenomyosis in general.
Thank you very much for your suggestions, we updated the table no 1 according to your suggestions
- Line 42: The diagnosis of adenomyosis by TVS is still difficult. The lines 42 – 46 suggest that EC in adenomyosis is a common finding. But the contrary is true. EC in patients with adenomyosis is rare. And there is no literature on the differentiation of transformation or coexistence by TVS. Thus, these lines could be discussed in the discussion section.
Thank you for this suggestion; I rewrote this paragraph (lines 42-47), emphasizing lack of pathognomic features of EC in adenomyotic uterus.
- Line 309: I agree. However, I think the difficulty to detect EC in patients with adenomyosis should be highlighted. The typical early uterine bleeding might be missing or might be misunderstood. The sonographic appearance of EC in an adenomyotic uterus might not be visible. What does this mean for clinical counselling? Maybe it should be discussed that this could also influence the decision making in patients with adenomyosis towards hysterectomy, especially in patients with hormone-replacement therapy or endocrine treatment of breast cancer
According to your suggestions chapter „conclusions” was enriched (lines 333-336)
Reviewer 3 Report
This review address a topic that is understimated but, need attention. Most of the time adenomyosis is considered just a reason for pain without risk for the woman. The article is well written with excellent tables and figures.
Author Response
Dear Reviewer,
Thank you very much for your appreciation. I hope that these review fill the gap in the knowledge about EC and its possible relationships with adenomyosis.
Regards,
Maria Szubert